# Comparison between Urine and Cervical High-Risk HPV Tests for Japanese Women with ASC-US

**DOI:** 10.3390/diagnostics11101895

**Published:** 2021-10-14

**Authors:** Hiroyuki Yamazaki, Tsuneyuki Wada, Hiroshi Asano, Hiromasa Fujita, Kazuhira Okamoto, Hidemichi Watari

**Affiliations:** 1Department of Obstetrics and Gynecology, Hokkaido University Graduate School of Medicine and Faculty of Medicine, Kita 15, Nishi 7, Kita-Ku, Sapporo 060-8638, Japan; cvm51396@elms.hokudai.ac.jp (H.Y.); asano.hj@gmail.com (H.A.); watarih@med.hokudai.ac.jp (H.W.); 2Hokkaido Cancer Society, 1–15, Kita 26, Higashi 14, Higashi-Ku, Sapporo 065-0026, Japan; tsunetsune0930@gmail.com (T.W.); fujita-hiromasa@hokkaido-taigan.jp (H.F.)

**Keywords:** uterine cervical neoplasms, human papillomavirus DNA tests, early detection of cancer, atypical squamous cells of the cervix, urinalysis

## Abstract

Most uterine cervical cancers are caused by the persistent infection of the high-risk human papillomavirus (hrHPV). Thus, the hrHPV-DNA test, which examines specimens from the cervix, is the standard screening method as well as cytology in western countries. Urine sampling for the hrHPV-DNA test would be easier and help improving screening rates. This study prospectively investigated the concordance between urine and cervical hrHPV tests for patients with atypical squamous cells of undetermined significance (ASC-US) in cervical cytology. We recruited 338 women with the cytologic diagnosis of ASC-US and performed hrHPV-DNA tests to both samples from the uterine cervix and first void urine, using the Cobas 4800 system. In all hrHPV genotypes, the simple concordance rate was 90.8% (307/338) and the Kappa statistic value was 0.765, which shows substantial concordance. The positive concordance rate was 70.5% (74/105), which was the rate excluding women who had negative results in both tests. When limited to types 16 and 18, the simple concordance rate was 98.8% (334/338), and the Kappa statistical value was calculated to be 0.840, which showed almost perfect concordance. The positive concordance rate resulted in 81.8% (18/22). We conclude that the urine hrHPV-DNA test could substitute the cervical test in women with ASC-US.

## 1. Introduction

Although uterine cervical cancer is the fourth most common malignancy among women and reported to have caused 311,000 deaths in 2018 over the world, cervical cancer could be eliminated, the World Health Organization announced, by spreading vaccination against the human papillomavirus (HPV), screening, and treatment for precancerous lesions [1]. Unfortunately, Japan is one of the developed countries with a higher incidence of cervical cancer: the age-standardized incidence in 2020 was 15.2 per 100,000 women, which is 2.45 times higher than that of the United States [2]. This results from the considerably poor organization of HPV vaccination and cervical cancer screening. The vaccination rate dropped to almost 0% after 2013 when the Japanese ministry suspended the proactive recommendation because of potential adverse events reports, most of which were finally found not to be related to vaccination [3]. Cervical cancer screening should play a more important role in earlier detections for women who failed to receive the vaccine at the appropriate ages. However, the screening is usually less widespread in vaccine-underdeveloped countries as well as in Japan [1,4]. The HPV-DNA test using a urine sample is expected to substitute the cervical test by including self-sampling because it is easier for examinees to collect specimens themselves, there is no need to prepare examination beds and expensive collection kits, and it has an advantage in preventing coronavirus transmission by reducing contact with the medical staff during the recent COVID-19 pandemic. Several reports have shown the good concordance between urine and cervical sampling to detect HPV infection in women with cervical intraepithelial neoplasia (CIN) [5,6,7,8,9]. Daponte et al. reviewed the literature and introduced a urine HPV test as a most promising tool to change cervical cancer prevention strategies [10]. However, few studies have revealed the feasibility of alternatives to cervical HPV testing in the general screening.

This study prospectively investigated the concordance between urine and cervical hrHPV test for patients with the cytologic diagnosis of ASC-US. We focused on the women with ASC-US in this study because they are the population with the highest implication for assessing the presence of HPV infection in the screening by cervical cytology.

## 2. Materials and Methods

### 2.1. Study Design

This was a prospective observational study to examine whether the results of the hrHPV-DNA test from urine were consistent with those from the cervix. We recruited the women who met the eligibility criteria described below, performed a cervical hrHPV-DNA test, colposcopy and collected the first void urine. We evaluated the concordance between them.

### 2.2. Subjects

This study was conducted from April 2016 to October 2020 for women who underwent cervical cytology as a cervical cancer screening test at our group institutions in each area of Hokkaido, Japan. Eligibility requirements were the women with ASC-US in the cervical cytology and those who visited and underwent cervical hrHPV-DNA test and colposcopy in the Sapporo Center, the Hokkaido Cancer Society. We excluded women who had a history of diagnosis and/or treatment of CIN or examination for cytological abnormality because they could be aware of their hrHPV status.

### 2.3. Sample Collection and Processing

Cervical samples were collected with Cervex-Brush (Rovers Medical Devices B.V., Oss, The Netherlands) and put the head of the brush into BD SurePath collection vial including 10 mL of BD SurePath liquid (BD, Becton Dickinson Ltd., Burlington, NC, USA). For the urine samples, participants themselves collected the first 20–50 mL void with a paper urine collection cup before a pelvic examination, investigators utilized 10 mL of the sample to extract DNA. After stirring, they were centrifuged at 3000 rpm (1800 rcf) for 10 min. The supernatant was removed, and 2 mL of BD SurePath liquid was added to the remaining pellet, which was performed on the day provided.

### 2.4. hrHPV-DNA Detection

Both cervical and urine samples were evaluated for hrHPV-DNA with Cobas 4800 system (Roche Molecular Diagnostics, Pleasanton, CA, USA), which features fully automated specimen preparation and subsequent real-time PCR procedure. The system detects total 14 genotypes of hrHPV-DNA (HPV-16, -18, -31, -33, -35, -39, -45, -51, -52, -56, -58, -59, -66, and -68), and reports results for HPV-16, HPV-18 and the other hrHPV types, respectively. The human cellular globin gene was also co-amplified to make a control for cell number adequacy, extraction, and amplification.

### 2.5. Clinical Examination and Pathological Analysis

Colposcopy and guided punch biopsy were performed by the gynecologist or the gynecologic oncologist, and all pathological findings were re-assessed according to WHO Classification of Tumors 5th Edition, Female Genital Tumors [11]. For women with CIN2 lesion but obtaining negative results from both cervical and urine hrHPV-DNA tests, additional immunohistochemistry (IHC) was carried out including p16 (Roche, Heidelberg, Germany) and Ki67 (Agilent Dako, Stockport, UK) on formalin-fixed, paraffin-embedded (FFPE) cervical tissue sections according to the manufacturer’s instructions.

### 2.6. Statistical Analysis

The results for hrHPV-DNA were defined according to the Cobas 4800 system reports: they are described as type 16/18 positive when either HPV-16 or HPV-18, or both are positive; overall hrHPV positive when either type 16/18 and other genotypes of hrHPV, or both are positive; negative hrHPV when neither type 16/18 or other genotypes are detected, which were equally applied to both cervical and urine samples. The concordance rate and kappa coefficient were analyzed using irr package on R: A language and environment for statistical computing (R Core Team, 2021. R Foundation for Statistical Computing, Vienna, Austria.). Frequency of HPV infection and sensitivities were evaluated with a chi-square test with R. The results were considered statistically significant at *p* < 0.05. Results of the Kappa statistical value saw the concordance considered as poor (<0), slight (0.01–0.20), fair (0.21–0.40), moderate (0.41–0.60), substantial (0.61–0.80) and almost perfect (0.81–1.00) [12].

## 3. Results

### 3.1. Study Population

From 2016 to 2020, a total of 148,342 women underwent cervical cytology as a cervical cancer screening test at our institutions. All specimens were evaluated at one testing facility, Sapporo center of the Hokkaido Cancer Society, and 1347 specimens were determined to be ASC-US. Of those 352 women who visited the center, 14 women who had a history of diagnosis and/or treatment of CIN or examination for cytological abnormality were excluded. We recruited a total of 338 women who met the eligibility requirements and agreed to participate in this study (the consent acquisition rate: 100%). The first void urine sample was collected before performing colposcopy-guided punch biopsy and cervical hrHPV-DNA testing. No samples were invalid for cell number adequacy, extraction, and amplification in Cobas 4800 system, and a total of 338 women were included. The median (range) age was 46 (20–67) years old. Guided punch biopsies turned out that one (0.3%) had adenocarcinoma, seventeen (5.0%) had CIN3 lesions including carcinoma in situ, and twenty-five (7.4%) had CIN2 lesions, shown in Figure 1.

### 3.2. Concordance between the Urine and Cervical Samples

Concordances between the urine and cervical samples were evaluated by two outcomes: overall hrHPV genotypes, shown in Table 1, and limited to HPV type 16 and 18, shown in Table 2. For overall hrHPV genotypes, the total concordance rate was 90.8% (307/338) and the Kappa statistic value was 0.765 (95%CI: 0.717–0.806), indicating substantial concordance. The positive concordance rate, which was the concordance rate excluding women who had negative results both in the cervical and urine tests, was 70.5% (74/105). This means that 74 of 105 women who had any positive results were positive both in the cervical and urine tests, whereas 31 women were positive only on one of the tests; 18 (17.1%) women were positive only in the cervical test, and 13 (12.4%) were positive only in the urine test.

Limiting the result to HPV 16 and 18, the total concordance rate was 98.8% (334/338) and the Kappa statistical value was calculated to be 0.894 (95%CI: 0.870–0.914), demonstrating almost perfect concordance. The positive concordance rate for HPV16 and 18 was 81.8% (18/22). Of the 22 women who had any positive result, two (9.1%) women were positive only in the cervical test and two (9.1%) in the urine test.

### 3.3. Sensitivities Compared by the Age Group

Table 3 and Table 4 show the positive rate of hrHPV by age group and CIN lesion in the cervical and urine test, respectively. A total of 92 women (27.2%) had positive results of the hrHPV in the cervical test, while 87 women (25.7%) had positive results in the urine test. The positive rate of hrHPV-DNA in the younger age group (<35 years old) was 41.7% (25/60) in the cervical test and 46.7% (28/60) in the urine test, which was significantly higher than that of the older age group (>35 years old) in both of the tests (41.7% vs. 24.1% in cervix, 46.7% vs. 21.2% in urine. *p* < 0.01, respectively). CIN lesions were diagnosed by colposcopy-guided punch biopsy in 111 women, and we defined CIN3+ lesions as CIN3 and invasive carcinoma, and CIN2+ lesions as CIN2, CIN3, and invasive carcinoma. A total of 18 women had CIN3+ lesions, and the prevalence rates by age group were 2.6%, 9.3%, 5.3%, and 4.7% for the 20s, 30s, 40s, and over 50s, respectively, while the prevalence rates of CIN2+ lesions by age group were 12.8%, 37.2%, 11.3%, and 4.7% for the 20s, 30s, 40s, and over 50s, respectively. We analyzed the sensitivities of the hrHPV test for each of CIN3+ and CIN2+ lesions by age group. The sensitivity for CIN3+ lesions was good in both the cervical and the urine test, which saw 16 (88.9%) and 15 (83.3%) women with CIN3+ lesions, respectively. Only one woman (5.6%) was negative for both the cervical and urine tests. The sensitivities to detect CIN2+ were 69.8% (30/43) in the cervical test and 65.1% (28/43) in the urine test. The sensitivities in the age group of the 20s were 40.0% (2/5) in both tests, while those of women over 30 years old were 73.7% (28/38) and 68.4% (26/38) in the cervical and the urine test, respectively.

### 3.4. CIN2+ with Negative hrHPV-DNA Test

No hrHPV-DNA was detected in both the cervical and the urine test in nine (20.9%) of the forty-two patients with CIN2+ lesions. Of the nine patients, one had CIN3 and the others had CIN2 lesions, and three underwent surgical treatment; the others received follow-up by the cervical cytology without any treatment. Of the six patients without treatment, five showed regression to NILM in the cervical cytology within a year, and the other had no disease progression. We conducted IHC of p16 and Ki67 for cervical lesions of the eight patients with CIN2. As shown in Table 5, three patients (# 1–3) showed positive staining for both p16 and Ki67, whereas five patients (# 4–8) were negative for p16 with positive expression of Ki67 except for #4. These additional findings did not affect the decision of treatment plans.

## 4. Discussion

We reported here excellent concordance between urine and cervical sampling in hrHPV-DNA tests in women with ASC-US. The concordance rates were 98.8% for type 16/18 and 90.8% for overall high-risk types, respectively. The previous reports also showed comparable concordances regardless of age, race, or region [5,6,7]. In Japan, the standard screening method is cervical cytology performed by a gynecologist every two years and the target receiving rate is set to 70%. Nevertheless, it was revealed that only 43.7% of women underwent gynecological screening in 2019, although 69.7% of women received general health checkups [4]. The shortage of gynecologists would be one of the causes impairing the participation rate, particularly in rural areas [13]. Though self-sampling kits demonstrated highly acceptable performance in usage and sensitivity, these were not widely accepted due to the need to use special equipment and the resistance of women to put the equipment into the vagina themselves in Japan [8,9]. The urine hrHPV-DNA test could be a substitute for the cervical test and is expected to increase receiving rate in the uterine cervical cancer screening because that does not need a special sampling kit, examination chair, or gynecologist examination.

Discussing the discordant cases who had different results for their urine and cervical samples, 5.3% of participants were negative in the urine test but positive in the cervix, shown in Table 1. They would be the additional false negatives when shifting to urine sampling from cervical hrHPV-DNA tests in cervical cancer screening. A few reports have also shown superior sensitivities in the urine test compared to the cervical sampling [5,6,7]. However, the discordance does not always imply a disadvantage of the urine test because 3.8% of participants were positive only in the urine test. They would be the false negatives when screened by the cervical hrHPV-DNA test and salvaged when shifting to the urine screening. This population was also shown by the previous reports, in which 5–10% of women were positive only in the urine test. One of the reasons why a certain percentage of women would be detected by either test is the sensitivity of a single sampling. The previous reports showed that a single sampling could not detect all HPV infections in the cervical test [14,15]. Multiple times of sampling would improve the concordance and sensitivity of the hrHPV-DNA test, and the urine test is more suitable for multiple times sampling because the examinees can collect the samples without gynecological examination nor special kits for self-sampling.

We also focused on the CIN2+ cases without hrHPV-DNA detections, shown in Table 4. These IHC will show higher expressions in more severe intraepithelial lesions; p16 indicates the interference of oncoprotein E7 from hrHPV types with pRB, and Ki67 is associated with cell proliferation, which suggests CIN progression [16]. Positive lesions of both p16 and Ki67 (# 1–3) agreed with the morphological diagnosis of CIN2. Negative hrHPV-DNA test results could be caused by infections of low- or intermediate-risk genotypes [17]. Because nobody underwent extensional HPV-DNA tests to detect any genotypes, we have no idea which genotypes of HPV were involved. No obvious positive lesions (# 4–8) were unmatched with morphological findings. CIN2 without hrHPV could be acceptable for false negatives because all six subjects who were observed without any treatment had no progression in our study. Moreover, Hosaka et al. reported the extremely low risk of two-years progression in CIN2 without hrHPV (0/165) [18]. Women who had CIN3+ lesions (# 9) could be a “victim” of hrHPV screening, which omits the minor risk of missing cancer or precancerous lesions without detectable hrHPV. The Cobas4800 system, used in this study, is estimated to detect 97.4% of hrHPV genotypes, which were detected among Japanese women with cervical cancer [19]. This suggests at least 2–3% of women with HPV infection would be classified as negative. However, it is expected to detect more lesions than cytological screening in total for the higher sensitivity in hrHPV-related lesions [20,21,22,23,24]. Co-testing of cytology and hrHPV-DNA might reduce false negatives but significantly increase false positives, resulting in overdiagnosis [25,26,27].

Before the introduction of urine sampling to the general screening for cervical cancer, we should answer the concerns regarding the excessive examination and treatment in areas where HPV vaccines are not widely spread, and we should establish algorithms to manage women with positive hrHPV-DNA but without CIN. We focused on the results of women with ASC-US in this study because they were expected to include all stages of HPV infections; cancer, CIN as a precancerous lesion and a cytological abnormality as a transient HPV infection, and HPV non-related lesions; CIN without HPV infection, inflammation, and squamous metaplasia. This diversity makes them the most beneficial population to detect hrHPV infections in cytological screening. The positive rates of hrHPV in ASC-US are reported to be 30–50% and for younger subjects are more likely to be positive. The prevalence of CIN2+ in ASC-US is reported to be 7–12% [28,29,30,31,32,33]. In our cases, the significantly higher positive rate of hrHPV test in <35 year old women was observed, while there was less prevalence for CIN2+. These false positives could be caused by transient HPV infections. Indeed, Kono et al. reported the relationship between the experimental introduction of the hrHPV test in screening and a significantly higher referral rate in Japan [34]. It is partially controversial to detect CIN2 lesions by cervical cancer screening because they include more transient lesions in vaccine-underdeveloped areas [19,23,35]. We did not restrict the detection target and analyzed the sensitivities for both CIN2+ and CIN3+.

Several limitations should be noted, which are the regionality and the small number of cases. Though this is a report of a prospective study, the participants were recruited when they received ASC-US results and were examined at a single institute in Sapporo, Japan. Furthermore, we did not compare HPV genotype but hrHPV. Further study would be needed to confirm the sensitivity in women with positive hrHPV lesions and CIN3+ by HPV genotypes and age groups because our study involved 68.9% of women who were negative for both cervical and urine tests.

In conclusion, we demonstrated the good concordance between urine self-sampling and cervical sampling by gynecologists for hrHPV-DNA tests in women with ASC-US. The urine hrHPV-DNA test could be a good alternative to the cervical hrHPV-DNA test in women who do not have the opportunity to receive cervical cancer screening. We need to add a reminder of the possibility that the introduction of the urine hrHPV-DNA test in the screening program would increase both the screening rate and false positives in vaccine-underdeveloped countries.

## Figures and Tables

**Figure 1 diagnostics-11-01895-f001:**
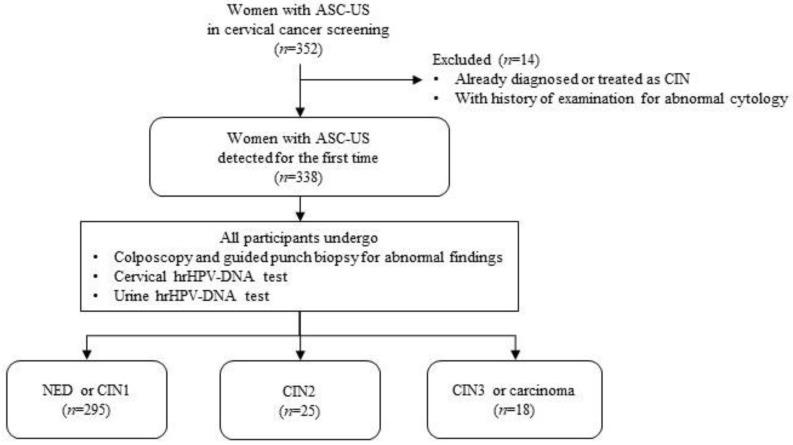
The flow of the study population. ASC-US, atypical squamous cells of undetermined significance; CIN, cervical intraepithelial neoplasm; hrHPV, high-risk HPV; NED, no evidence of disease.

**Table 1 diagnostics-11-01895-t001:** Agreement of high-risk HPV-DNA tests in overall genotypes.

		Urine Test	
		Positive	Negative
Cervical test	Positive	74(21.9%)	18(5.3%)
	Negative	13(3.8%)	233(68.9%)

The percentages indicate the ratios to the total number of participants.

**Table 2 diagnostics-11-01895-t002:** Agreement of high-risk HPV-DNA tests in types 16 and 18.

		Urine Test	
		Positive	Negative
Cervical test	Positive	18(5.3%)	2(5.9%)
	Negative	2(5.9%)	316(93.5%)

The percentages indicate the ratios to the total number of participants.

**Table 3 diagnostics-11-01895-t003:** The positive rate of hrHPV in the cervical test by age group and CIN lesion.

Age	Total	CIN1	CIN2	CIN3+
total	27.2% (92/338)	32.4% (22/68)	56.0% (14/25)	88.9% (16/18)
20–34	41.7% (25/60)	42.1% (8/19)	50.0% (5/10)	100% (3/3)
35-	24.1% (67/278)	28.6% (14/49)	60.0% (9/15)	86.7% (13/15)
20–29	35.9% (14/39)	46.7% (7/15)	25.0% (1/4)	100% (1/1)
30–39	44.2% (19/43)	22.2% (2/9)	66.7% (8/12)	100% (4/4)
40–49	24.7% (37/150)	38.7% (12/31)	55.6% (5/9)	75.0% (6/8)
50-	20.8% (22/106)	76.9% (1/13)	0% (0/0)	100% (5/5)

CIN, cervical intraepithelial neoplasm; CIN3+, at least CIN3.

**Table 4 diagnostics-11-01895-t004:** The positive rate of hrHPV in the urine test by age group and CIN lesion.

Age	Total	CIN1	CIN2	CIN3+
total	25.7% (87/338)	30.9% (21/68)	52.0% (13/25)	83.3% (15/18)
20–34	46.7% (28/60)	52.6% (10/19)	50.0% (5/10)	100% (3/3)
35-	21.2% (59/278)	22.4% (11/49)	53.3% (8/15)	80.0% (12/15)
20–29	43.6% (17/39)	60.0% (9/15)	25.0% (1/4)	100% (1/1)
30–39	39.5% (17/43)	11.1% (1/9)	58.3% (7/12)	100% (4/4)
40–49	24.0% (36/150)	35.5% (11/31)	55.6% (5/9)	62.5% (5/8)
50-	16.0% (17/106)	0% (0/13)	0% (0/0)	100% (5/5)

CIN, cervical intraepithelial neoplasm; CIN3+, at least CIN3.

**Table 5 diagnostics-11-01895-t005:** Immunohistochemistry for p16 and Ki67 on high-risk HPV negative patients with CIN2.

No.	Diagnosis by Punch Biopsy	Age	p16 IHC	Ki67 IHC	Other Findings	Outcome
1	CIN2	45	positive in all layers cells	positive in the lower two-thirds of the epithelium	mitotic cells, cell atypia	treated with conization
2		41	partially positive	partially positive in basal cells		cytologic abnormality (at most ASC-US)
3		37	positive in parabasal cells	weakly positive in parabasal cells	koilocytosis, inflammatory cell infiltration	normal cytology
4		33	negative	partially positive in basal cells		treated with vaporization
5		21	negative	negative		normal cytology
6		30	negative	negative		normal cytology
7		26	negative	negative		normal cytology
8		26	negative	negative	inflammatory cell infiltration	normal cytology
9	CIN3	48	untested	untested	mitotic cells, cell atypia, koilocytosis	treated with conization

CIN, cervical intraepithelial neoplasm; IHC, immunohistochemistry; ASC-US, atypical squamous cells of undetermined significance.

## Data Availability

The data are not publicly available due to the study design and not being informed to participants.

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
