# Peer review of "Comparison between Urine and Cervical High-Risk HPV Tests for Japanese Women with ASC-US"

_diagnostics, 2021, doi:10.3390/diagnostics11101895_

Round 1

Reviewer 1 Report

RE: Comparison between urine and cervical high-risk HPV tests for Japanese women with ASC-US (diagnostics-1398299)

The study by Hiroyuki Yamazaki et al. aims at evaluating the validity/consistence of urine and cervical hrHPV samples in women referred with ASCUS and reports an excellent concordance and Kappa values between urine and cervical sampling of 98.8% for types 16/18 (Kappa statistical value was calculated to be 0.894) and 90.8% for overall high-risk types (Kappa statistic value was 0.765) respectively using COBAS.

Nowadays, in the era where cytolological screening of cervical cancer moves on to biomolecular hrHPV detection (either cervical – selfsampled - urine hrHPV tests) in terms of increasing screened numbers of the population, studies proving the validity of the new methodologies such as urine samples, like the present one, remain necessary, crucial and helpful until these approaches become established and introduced globally.

Please add the following recent review as reference

Daponte, A.; Michail, G.; Daponte, A.-I.; Daponte, N.; Valasoulis, G. Urine HPV in the Context of Genital and Cervical Cancer Screening—An Update of Current Literature. Cancers 2021, 13, 1640. https://doi.org/10.3390/cancers13071640

Author Response

Reviewer1

Dear Reviewer1

We would like to thank you for your encouraging comments on our manuscript. Your comment has been helpful in our revision of the manuscript.

Comments and Suggestions for Authors

RE: Comparison between urine and cervical high-risk HPV tests for Japanese women with ASC-US (diagnostics-1398299)

The study by Hiroyuki Yamazaki et al. aims at evaluating the validity/consistence of urine and cervical hrHPV samples in women referred with ASCUS and reports an excellent concordance and Kappa values between urine and cervical sampling of 98.8% for types 16/18 (Kappa statistical value was calculated to be 0.894) and 90.8% for overall high-risk types (Kappa statistic value was 0.765) respectively using COBAS.

Nowadays, in the era where cytolological screening of cervical cancer moves on to biomolecular hrHPV detection (either cervical – selfsampled - urine hrHPV tests) in terms of increasing screened numbers of the population, studies proving the validity of the new methodologies such as urine samples, like the present one, remain necessary, crucial and helpful until these approaches become established and introduced globally.

Please add the following recent review as reference

Daponte, A.; Michail, G.; Daponte, A.-I.; Daponte, N.; Valasoulis, G. Urine HPV in the Context of Genital and Cervical Cancer Screening—An Update of Current Literature. Cancers 202113, 1640. https://doi.org/10.3390/cancers13071640

We added a sentence in Introduction referring to his review article and References.

Page 2, Line52-53:

Daponte et al. reviewed literatures and introduced as a most promising tool, which could change the cervical cancer prevention strategies. [10]

Page 9, Line 316-317:

Daponte, A.; Michail, G.; Daponte, A.-I.; Daponte, N.; Valasoulis, G. Urine HPV in the Context of Genital and Cervical Cancer Screening—An Update of Current Literature. Cancers 2021, 13, 1640, doi:10.3390/cancers13071640.

Reviewer 2 Report

  1. It is considered valuable as a research that studies the clinical significance of HPV DNA test using urine that is easy to sample. This will be an opportunity to prevent cervical cancer for women who are excluded from cervical screening.
  2. It is considered meaningful to check the usefulness of a urine sample by comparing HPV DNA tests of biopsies, cervix, and urine samples for ASCUS patients. However, it is not advisable to conclude that the use of a urine sample can replace a cervical sample in ASCUS patients.
  3. It is judged that the table and figure need some correction.
     1) Figure 1
       - Full names of abbreviations such as ASCUS, NED, and CIN must be explained.
       - Why is only the HPV DNA result for CIN 2 (n=25) written in the last square? There is no explanation in the text.
    2) Table 1, 2: It seems that the table is displayed like a figure. It would be good to create a table by aligning all the letters horizontally.
    3) Table 3
      - Divide CIN1, CIN2, and CIN3 to create an analysis table for the results, so it will be easier to understand.
     - It is not clear why CIN2 (25 cases) and CIN3 (18 cases) are added to 43 CIN2+ class.
  4. Although the urine HPV DNA test is a meaningful result to be used as a screening diagnosis, a specific comparative analysis of HPV genotype is needed if it is intended to replace a cervical specimen. Therefore, it is necessary to accurately point out the limitations of this study and suggest future improvements, and it is more preferable to conclude that it suggests the possibility of clinical application of urine samples.

Author Response

Reviewer2

Dear Reviewer

We would like to thank you for your encouraging comments on our manuscript. Your comment has been helpful in our revision of the manuscript. We described the point-by-point responses to your comments.

Comments and Suggestions for Authors

  1. It is considered valuable as a research that studies the clinical significance of HPV DNA test using urine that is easy to sample. This will be an opportunity to prevent cervical cancer for women who are excluded from cervical screening.

Thanks for your comment. We are trying to create a stir in the strategy for preventing cervical cancer.

2. It is considered meaningful to check the usefulness of a urine sample by comparing HPV DNA tests of biopsies, cervix, and urine samples for ASCUS patients. However, it is not advisable to conclude that the use of a urine sample can replace a cervical sample in ASCUS patients.

We agree with your concern. Urine HPV-DNA tests indeed demonstrate slightly worse sensitivities than cervical sampling in previous studies. We revised conclusions according to our results.

Page 7, Line 270-273:

In conclusion, we demonstrated the good concordance between urine self-sampling and cervical sampling by gynecologists for hrHPV-DNA test in women with ASC-US. The urine hrHPV-DNA test could be a good alternative to the cervical hrHPV-DNA test in women away from the opportunity to receive cervical cancer screening.

3. It is judged that the table and figure need some correction.
 1) Figure 1
   - Full names of abbreviations such as ASCUS, NED, and CIN must be explained.
   - Why is only the HPV DNA result for CIN 2 (n=25) written in the last square? There is no explanation in the text.

We revised Figure 1 according to your comments.

Page 3, Figure 1: Changed

Page 3, Line 128-130 (Figure 1, abbreviations):

ASCUS, atypical squamous cells of undetermined significance; CIN, cervical intraepithelial neo-plasm; hrHPV, high-risk HPV; NED, no evidence of disease

2) Table 1, 2: It seems that the table is displayed like a figure. It would be good to create a table by aligning all the letters horizontally.

We revised Table 1 and 2 according to your comments.

Page 4, Table 1: Changed

Page 4, Table 2: Changed

3) Table 3
  - Divide CIN1, CIN2, and CIN3 to create an analysis table for the results, so it will be easier to understand.
We divided Table 3 into new Table 3 and Table 4, in which we described the positive rate of hrHPV resulted from cervical test and urine test, respectively. We added several sentences to 3.3. Sensitivities compared by the age group and Discussion.

Page 5, Line 153-155:

Tables 3 and 4 showed the positive rate of hrHPV by age group and CIN lesion in the cervical and urine test, respectively. A total of 92 women (27.2%) had positive results of the hrHPV in the cervical test, while 87 women (25.7%) in the urine test.

Page 5, Line 160-169:

CIN lesions were diagnosed by colposcopy-guided punch biopsy in 111 women, and we defined CIN3+ lesions as CIN3 and invasive carcinoma, and CIN2+ lesions as CIN2, CIN3, and invasive carcinoma. A total of 18 women had CIN3+ lesions, and the prevalence rates by age group were 2.6%, 9.3%, 5.3%, 4.7% for the 20s, 30s, 40s, over 50s, respectively, while the prevalence rates of CIN2+ lesions by age group were 12.8%, 37.2%, 11.3%, 4.7% for the 20s, 30s, 40s, over 50s, respectively. We analyzed the sensitivities of the hrHPV test for each of CIN3+ and CIN2+ lesions by age group. The sensitivity for CIN3+ lesions was good in both the cervical and the urine test, which saw 16 (88.9%) and 15 (83.3%) women with CIN3+ lesions, respectively. Only one woman (5.6%) was negative for both the cervical and urine tests.

Page 5: Table 3 changed.

Table 3. The positive rate of hrHPV in the cervical test by age group and CIN lesion.

Page 5, Table3 (footnote):

CIN, cervical intraepithelial neoplasm; CIN3+, at least CIN3

Page 5: Table 4 added.

Table 4. The positive rate of hrHPV in the urine test by age group and CIN lesion.

Page 5, Line 180; Page 6, Line 183:

Table 5

 - It is not clear why CIN2 (25 cases) and CIN3 (18 cases) are added to 43 CIN2+ class.

Thank you for your comment. We did not decide the target to detect by a urine test because the urine HPV test is not a standard method yet. The sensitivity for CIN3+ is one of the most critical indicators to be an alternative to cervical tests. At the same time, CIN2+ is representative target of general cervical cancer screening, and we expected that more transient lesions would be observed in CIN2+. We added an explanation on page 8 to clarify the reasons.

Page 8, Line 257-260:

It is partially controversial to detect CIN2 lesions by cervical cancer screening because they include more transient lesions in vaccine-underdeveloped areas. [19,23,35] We did not restrict the detection target and analyzed the sensitivities for both CIN2+ and CIN3+.

Page 10, Line 387-389:

  1. Cuzick, J.; Arbyn, M.; Sankaranarayanan, R.; Tsu, V.; Ronco, G.; Mayrand, M.-H.; Dillner, J.; Meijer, C.J.L.M. Overview of Human Papillomavirus-Based and Other Novel Options for Cervical Cancer Screening in Developed and Developing Countries. Vaccine 2008, 26, K29-K41, doi:10.1016/j.vaccine.2008.06.019.

4. Although the urine HPV DNA test is a meaningful result to be used as a screening diagnosis, a specific comparative analysis of HPV genotype is needed if it is intended to replace a cervical specimen. Therefore, it is necessary to accurately point out the limitations of this study and suggest future improvements, and it is more preferable to conclude that it suggests the possibility of clinical application of urine samples.

Thank you for your comment, and we added limitations about HPV genotype analysis on page 8. We agree with your suggestion, and we would like to perform a specific comparative analysis of HPV genotype in a future study.

Page7-8, Line 262-266

Furthermore, we did not compare HPV genotype but hrHPV. Further study would be needed to confirm the sensitivity in women with positive hrHPV lesions and CIN3+ by HPV genotypes and age groups because our study involved 68.9% of women who were negative for both cervical and urine tests.